# The Degradation and Aging of Biological Systems as a Process of Information Loss and Entropy Increase

**DOI:** 10.3390/e25071067

**Published:** 2023-07-15

**Authors:** Vladimir V. Aristov, Alexey V. Karnaukhov, Anatoly S. Buchelnikov, Vladimir F. Levchenko, Yury D. Nechipurenko

**Affiliations:** 1Federal Research Center “Computer Science and Control” of Russian Academy of Sciences, Vavilova Str. 40, 119333 Moscow, Russia; aristovvl@yandex.ru; 2Institute of Cell Biophysics of the Russian Academy of Sciences, Moskovskaya obl., Institutskaya Str. 3, 142290 Pushchino, Russia; alexeykarnaukhov@yandex.ru; 3Laboratory of Molecular and Cellular Biophysics, Sevastopol State University, Universitetskaya Str. 33, 299053 Sevastopol, Russia; tolybas@rambler.ru; 4Laboratory of DNA–Protein Interactions, Engelhardt Institute of Molecular Biology of Russian Academy of Sciences, Vavilova Str. 32, 119991 Moscow, Russia; 5Sechenov Institute of Evolutionary Physiology and Biochemistry of the Russian Academy of Sciences, Russian Federation, Thorez 44, 194223 St. Petersburg, Russia; lew@lew.spb.org

**Keywords:** aging of biological systems, statistical entropy, Schrödinger’s concepts, negentropy feeding, Kullback–Leibler entropy, skin age

## Abstract

The problem of the degradation and aging of bioorganisms is herein considered from the viewpoint of statistical physics. Two typical timescales in biological systems—the time of metabolic processes and the time of the life cycle—are used. A kinetic equation describing the small timescales of the systems’ characteristic processes in is proposed. Maintaining a biosystem in a time-stable state requires a constant inflow of negative entropy (negentropy). Ratios are proposed to evaluate the aging and degradation of systems in terms of entropy. As an example, the aging of the epithelium is studied. The connection of our approach to the information theory of aging is discussed, as well as theoretical constructions related to the concept of cooperon and its changing with time.

## 1. Introduction

There are many views on the aging of biosystems; according to some estimates, there are about a hundred such theoretical constructs (see, for example, [1,2,3]). All aging theories can be divided into two groups, in the following way: (1) *theories of programmed aging*: apoptosis, etc.; and (2) *theories of aging as a consequence of error accumulation*, i.e., an increase in entropy due to different biological processes. Herein, we will consider this second type of theory.

According to Sinclair (see [4] and the corresponding references therein), it is possible to classify interpretations of aging into two types: evolutionary theories that try to explain why aging exists as a phenomenon, and biophysical and biochemical theories that seek to describe the direct cause of aging or age-related decline in function. Perhaps the first type of theory can be correlated with the interpretation of aging as a programmed process developed during evolution. The second type of theory corresponds to the “entropic” approach, which we develop here.

We treat aging as a physical phenomenon within the framework of the approach to the problem of describing the degradation of systems. This study can be seen as a continuation of our work that uses the concept of entropy to describe processes in biosystems [5]. Note that here, we follow the Schrödinger principle, which implies that a biological system in a time-stable state requires a constant influx of negative entropy (negentropy); however, in addition, in this paper, we consider changes in entropy itself in the body due to slow degradation processes.

An overview of the application of the entropy concept in biological systems is presented in [6] (see also [7,8,9,10]). Aging in all systems (both closed and open) means a transition on different timescales from a more nonequilibrium to a less nonequilibrium (with higher entropy) state, which actually follows the second principle of thermodynamics. Thus, we can combine theories of the aging of biosystems with some or other mechanisms of degradation, assuming the introduction of a general physical description. In addition, it is useful to review works [11,12,13,14,15,16].

One of the goals of this paper is to define a measure of the aging of biosystems, or in other words to introduce a quantitative assessment of the age of such systems. We will use the value of entropy, calculated using various approaches, as an estimate. The notion of statistical entropy in general and combinatorial entropy in particular will be considered. Using the example of changes in epithelial structure, we will show how Kullback–Leibler entropy grows. Variants of Karnaukhov’s [17,18,19,20,21,22] and Sinclair’s [23,24] information theories of aging will also be discussed (for a common issue, see [25]). These representations will hopefully allow us to develop a new approach to the experimental estimation of the biological age of a system.

## 2. Results

### 2.1. Degradation Processes in an Open Nonequilibrium System

In this paper, degradation is interpreted not as a violation of the “right” mechanism, but as a manifestation of general thermodynamic principles applied to open nonequilibrium structures and leading them to an equilibrium state. Because of such a process, the structure of the biosystem necessary for its normal functioning is disturbed (note that we can consider degradation as a weakening of the coordination and consistency of separate parts of the organism). The processes occurring in the organism can be divided into two types, which differ significantly in timescale. The first type, with characteristic time *τ*, corresponds to metabolism; the intensity of energy metabolism is conveniently estimated by the rate of oxygen consumption [26]. It is understood that aerobic organisms are considered. Accordingly, it can be estimated by the rate of glucose metabolism (see also [27,28,29]). Note that in [5], the kinetic model of inhomogeneous spatial relaxation describes this process. The second process is a slow degradation with a characteristic time *T* = *Nτ*, where *N* is a very large number, *τ* is the time of a characteristic metabolic reaction, and *T* is the average lifetime of the animal.

Times *T* and *τ* vary considerably, but the ratio of the average lifetime of an organism to the characteristic metabolic time is in some approximations a constant value, at least within a particular class of animals. For example, for mammals, an estimate of *T*/*t_H_* ~ 10^9^ is known, where *t_H_* is the time of one heartbeat. This estimate is “universal” in nature (it is equal to approximately the average number of heartbeats during a lifetime (except in humans, where it is 2–3 times bigger). For other classes of animals, this value may be different; for example, it is not quite correct to compare birds and mammals in this sense (see [26]). The heartbeat time *t_H_* and the characteristic metabolic time *τ* for mammals are related by a multiplier approximately equal to 30: the time of one mouse heartbeat *t_H_* = 0.1 s, and the time *τ* = 3 s, one elephant heartbeat *t_H_* = 2 s, and the time *τ* = 60 s, so that we obtain an estimate of the indicated ratio *N* = *T*/*τ* ~ 3 × 10^7^.

Let us consider the process of degradation using the example of human skin. It is known that human skin consists of several heterogeneous layers, among which the one of interest to us is a relatively homogeneous upper layer, the epidermis. In its turn, the epidermis also consists of several layers of epithelial cells, i.e., it has some thickness. This thickness varies with age; for example, a baby’s skin is thin, and with time, the skin thickens, reaching a maximum at a young age; then, it thins again, coarsens, and wrinkles appear in it. The cells of the lower basal layer of the epidermis—keratinocytes—rise upwards during differentiation, gradually reaching the upper (outer) stratum corneum, and then die and peel off. The differentiated keratinocytes are replaced by new cells in the basal layer, and the process is repeated many times throughout the human life.

We correlate cell aging with growth of its total entropy; maximum entropy corresponds to minimum cell viability. Here, we consider viability as the number of metabolic cycles the cell can survive. Thus, zero viability refers to a cell that has lost its ability to metabolize, is in the upper corneous layer of the epidermis, and can peel off from the surface at any moment.

Let us consider two periods of human life, young and old. At a young age, the viability of basal layer keratinocytes is higher than in an elderly person. Figure 1 schematically shows the dependence of cell entropy on their coordinates in epidermis thickness. The zero mark corresponds to the greatest depth, and the *L* mark to the surface. The green line shows such a dependence for the skin cells of a young person, and the blue line for the skin cells of an elderly person.

In work [5], we estimated the local statistical entropy and found that during aging, relaxation processes dominate over advection (transport) processes.

### 2.2. A Mathematical Description of Tissue Structure Disturbance and Entropy Change

Let us introduce the notion of a distribution function for a biological system. In the simplest case of a physical system, this function defines a dependence of particle density *f* (*x*, *ξ*) in the system on *x* coordinate and velocity *ξ*. It is shown in [5] that for the distribution function on the relaxation timescale, neglecting slow changes in aging, the stationary kinetic equation can be written as follows:(1)$$\xi \frac{\partial f}{\partial x}=\frac{1}{\tau }\left(f_{M}-f\right)$$
where *f_M_* is the equilibrium distribution function, and *f* is in general a nonequilibrium distribution function.

The traditional statistical Boltzmann entropy used for description of the spatial nonequilibrium states is the following moment:S=−∫flogf dξ

Profiles of the equilibrium and nonequilibrium local entropies are shown in Figure 7 in [5], illustrating the difference between the nonequilibrium (green bottom line, a “line of life”) and the equilibrium (blue upper line, a “line of death”) entropies with the same density, velocity, and temperature computed at any local spatial point. In [5], “It is seen that the local nonequilibrium entropy is less than the local equilibrium entropy”. The epithelium aging line corresponds to greater local entropy. According to the notations in Figure 1, the blue line corresponds to entropy in old age, and the green line to entropy in young age. The difference between these quantities, integrated over space, can give a characteristic of the age of the body’s skin:(2)$$\Delta S=\int_{0}^{L_{\mathrm{old}}}\left(S_{\mathrm{old}}-S_{\mathrm{young}}\right)dx$$

The difference under the integral can be calculated using the distribution function of the kinetic formulation, as mentioned above.

The question arises: in what simple way can entropy be used to mark the age of an organism, organ, or system of cells? There are various definitions of entropy. Calculating a distribution function is possible in principle, but rather complicated. We can propose a Kullback–Leibler entropy calculation for a model situation in which we describe the epithelium.

Using an analogy with the particle distribution function, we can introduce into consideration (on a different scale) a cell distribution function, which in a simple case can be described as cell density, depending on the coordinate we choose; this may be, for example, the density of skin epithelium cells *p*(*x*) on the surface along coordinate *x*. This density changes with time; some cells die off, others replace them, some cells are lost irretrievably, and hollows and wrinkles form on these skin areas.

To estimate epithelial aging, we can introduce the Kullback–Leibler divergence (relative entropy), which is given in the following form:(3)$$D\left(p∥q\right)=\sum_{x\in X}p\left(x\right)\log\frac{p\left(x\right)}{q\left(x\right)}$$Here, *p*(*x*) is the distribution (density) of the cells at the beginning of consideration, and *q*(*x*) is the distribution at the end. Relative entropy determines the closeness of the two distributions *p*(*x*) and *q*(*x*). We can consider the base distribution *p*(*x*) for young age. The distribution *q*(*x*) refers to older age. We construct below an illustrative example of a possible estimate.

Relative entropy grows in time, and Equation (3) takes into account, in fact, the loss of information when the initial distribution *p* is replaced by the final distribution *q*. The degradation time is determined by an invariant, which for mammals, as stated above, is of the order of 10^9^ (the ratio *T*/*τ*). Accordingly, we can then roughly estimate the value of $D\left(p∥q\right)$, assuming that *p*(*x*)/*q*(*x*) ~ 10^9^, i.e., log (*p*/*q*) ~ 9.

For illustration, the Kullback–Leibler entropy can be used to describe processes in the epidermis. Our estimates are theoretical, but we hope that experimental studies can be performed in the future. Age can be related to the value of the Kullback–Leibler entropy. Thus, three divergence values can be attributed to three different ages, and estimates can be made accordingly.

Chronic inflammation is one of the main biological markers of aging and elements of age-related disease pathogenesis. In this case, immune response processes demonstrate increasing chaos and mismatch with age [30]. This process can be intuitively associated with an increase in entropy in a biological system. However, it is also possible to give some quantitative assessment of the entropy increase (as well as the aging of an organism) if the concept of combinatorial entropy [6] is used correctly (see Appendix B). There is also an increase in autoimmune response, one of the causes of which is the accumulation of genetic errors in cellular genomes, which reflects the degradation of the genetic information of the organism. An example of such a phenomenon is chronic inflammation. In the future, we plan to build a specific simulation model in which the processes of aging, system entropy growth, and the degradation of genetic information will be interrelated.

Arrhythmia with age (as compared to a regular rhythm at a relatively young age) is characterized by an increase in statistical entropy. The distribution with a regular (referred to as sinus) rhythm has a knowingly smaller statistical entropy than the distribution with an irregular (chaotic) rhythm in arrhythmia.

A biological organism is a self-supported structure which retains its properties due to metabolism. However, metabolism on different spatial and temporal scales leads both to continuous reproduction of the structure, its restoration, and to violation of the integrity of the structure because of random factors. Thus, it is possible to describe telomere shortening, changes in tissue structure during cell replacement, clot growth, etc., in the model. Below is an example of epithelium aging.

Let us consider a change in the distribution function in the simplest case, when it is represented as a dependence of the cells’ density *p*(*x*) on their position (for example, if we take a skin area, draw a line, and describe the skin relief, we obtain a graph of such a function).

Let us divide the skin area into identical elements (“compartments”). Let the acts of metabolism in each of them produce a change in the value of *p*(*x*) in a random way. The “cell columns” above each “compartment” turn out to be of different height, but on average, their number decreases with age, as written above: *L*_old_ < *L*_young_ (Figure 1).

We performed a computer simulation of the aging process of the epithelium. We divided the skin area into 40 “compartments”, in each of which two processes took place: (1) a fast, chaotic one, simulating many acts of metabolism in which the density *p*(*x*) changed randomly; and (2) a slow one, which is associated with skin thinning with age.

Figure 2 shows the change in the skin structure on the surface because of such a simulation: first is a smooth structure, corresponding to the initial “young” age with the distribution *p*(*x*); next is a non-smooth structure of “older” age with the distribution *q*(*x*).

In Figure 1 and Figure 2, we can see chaotization in the disturbance of skin smoothness, which in the first case is represented as entropy fluctuations, and in the second case as changes in skin thickness in different skin areas. This chaotization corresponds to the growth of Kullback–Leibler entropy and the growth of statistical entropy in the movement from *S*_young_ (young age) to *S*_old_ (old age) in Figure 1. Using Kullback–Leibler entropy, *p*(*x*) can be regarded as the baseline distribution of skin density at a young age, and *q*(*x*) as the distribution at an older age.

Our approach allows us to introduce a definition of a measure of the aging of a biosystem. It is possible to quantify the age of the system using the entropy value calculated in Equation (2) or Equation (3); this will make it possible to estimate the biological age of the system.

In addition, as was already mentioned in the previous section, it is possible to compare this entropy with the classical statistical entropy used in our work [5]. With time, the statistical entropy grows, each part becomes more autonomous, there is chaotization in each part, and the combinatorial entropy increases (see Appendix B). With such “chaotization” and the disturbance of connections between parts of the system, the “smoothness” of the skin is disturbed.

The above Formula (3) for the Kullback–Leibler entropy allows us to estimate the increase in entropy when simulating skin aging. We simulated changes in the skin structure and calculated the corresponding Kullback–Leibler entropy as a function of the number of iterations (see Figure 3).

Figure 3b shows how the modulus of the difference between the Kullback–Leibler entropy and the unit changes as iterations are performed. This quantity, although with known fluctuations, increases, which characterizes the aging of the system. Such calculations can be made for the configuration of cells in some organ, or in the organism as a whole. At a young age, a certain configuration is fixed. At a later age, it changes (while the cells are repeatedly replaced); the comparison of the mentioned profiles will give an estimate of entropy.

The distribution change with time presented in Figure 2 shows the “stochastization” of spatial regions, which is equivalent to the greater independence of one area from another (decrease in cooperativity); thus, one can expect an increase in the calculated combinatorial entropy as well (see Appendix B).

### 2.3. The Development of Information Theories of Aging

The connection between the degradation of biological systems, processes of information loss, and growth of entropy in such systems inevitably leads to the idea of correlating the phenomenon of aging with the process of information loss. A.V. Karnaukhov [17,18,19,20,21,22] first introduced the term “informational theory of aging”. However, the key idea underlying this theory, which connects the phenomenon of aging with the accumulation of damage in the genome of the cells of a multicellular organism, has been expressed before. Here, we can mention, for example, the theory of somatic mutations proposed by L. Szilard [31,32].

In [17], an imitation model of aging for the individual and a similar species-specific model of “non-aging” are considered. The informational hypothesis assumes that aging in eukaryotes occurs due to informational degradation of cellular genetic material (the accumulation of random errors—or unrepaired mutations—in the genome). The phenomenon of chromosomes crossing over during gametogenesis and the subsequent competitive selection of gametes involved in the formation of the genetic material of descendants are considered the main mechanisms of the “rejuvenation” of genetic material (i.e., the reduction of the number of errors in the genome), during its transition from parental individuals to descendants. It is shown within the imitation model that this mechanism, existing in the vast majority of eukaryotes, ensures the stability of the gene pool of the population during an unlimited number of generations, while for a single organism, the number of errors (damages) in the genome of each cell of the organism increases with age. Crossing over, the process of the exchange of homologous regions between two copies of the genome at one of the stages of meiosis, ensures the random redistribution of genomic errors between gametes. An important property of this process is the significantly different from zero probability of gametes with a lower density of information errors. An information error in the genome is understood as an unrepaired mutation, such as a single nucleotide substitution.

Another important statement of the “informational aging” hypothesis is the assumption that gametes containing fewer informational lesions (errors) have a significantly higher chance of “generating” a new diploid organism than gametes whose genome contains a greater number of errors.

Note that most eukaryotic organisms produce a huge number of gametes, and this ensures a high level of competition between them, as well as a very strict selection. For example, in humans, tens of millions of sperm cells participate in the fertilization process of a single egg. Among spermatozoa, the most obvious (but not the only) traits, according to which the selection is made, are activity and speed of movement. About three million immature oocytes are produced in girls, even before birth. Only about three hundred reach the mature oocyte stage. Selection is based on different traits. The most important, but not the only trait, is the growth rate of the oocyte. After fertilization, the zygote and the blastomers emerge. There is also selection, which results in only a small percentage of the already fertilized eggs reaching birth.

Subsequently, the informational hypothesis received a direct independent confirmation; it was established using sequencing methods that the genome of the human germ cell line accumulates about two genetic errors per year [33]. In this connection, the authors of the aging information hypothesis considered it possible to use the term “theory” in the future. Besides, in experiments on laboratory animals (mice) it was shown that bone marrow transplantation from young syngeneic donors (within the inbred line) significantly increases both the average [18,19,20] and the maximum [22] lifespan of the recipient animals. This could also be considered a confirmation of this hypothesis.

It should be noted that a more detailed model of aging [21] of multicellular organisms was considered within the framework of the information theory. The main attention was paid to the analysis of the mutual influence of processes occurring at the level of individual organs and systems of the whole organism. Here, the basic aging scheme proposed within the framework of the information theory makes it possible to analyze in more detail the aging of highly organized living beings, whose specific feature is the complex structure of the organism, consisting of a large number of organs and systems, each of which may have its own aging rate. In this case, the unsynchronized aging of different organs and systems can affect the overall functionality of the organism in a very complex way.

Another variant of the “informational aging” theory, the so-called “Information theory of aging”, is developed by D. Sinclair and other authors [23,24,34,35,36,37]. These works also suggest that the main cause of aging is the “loss of information”. The authors proceed from the fact that there are two types of information in biological objects: conditionally speaking, digital or discrete information. These comprise the sequence of nucleotides in the DNA and analogue epigenetic information, or the epigenome. The information encoded in the sequence of nucleotides in DNA is stable and changes little in time. However, the way in which genes express themselves (“expressed”) is determined by the epigenome, a set of conditions and specific tags that regulate gene activity but do not affect the primary nucleotide structure of the DNA. If the genome is taken as a computer (hardware), then the epigenome can be considered computer software (software).

According to Waddington’s concept, with the same genome, a phenotype can be different due to different conditions of organism development, and may therefore have a different final epigenetic landscape (it should be noted that changes in this landscape with repeated repetitions can still be fixed in evolutionary transformations) [38]. The epigenome, due to its specific nature, is a much less stable way to store and transmit information; according to Sinclair, change in and loss of information in the epigenome leads to aging.

## 3. Discussion

A study of degradation on a large timescale *T* can be carried out on the basis of the above distribution function *f*. But it is also possible to introduce a specifically defined structural distribution function *f_T_* to describe structural changes in the system, indicating in what sense the biosystem is closed on such a large timescale.

Let us try to explain qualitatively the notion of a structural distribution function. Consider the interposition in the system of some real biological elements, which could be, for example, cells, isolated cellular subsystems, or organs (in a less detailed “partitioning” of the system). The implication is that in each element, there is a constant change of molecules as a result of metabolism, but the identity of each element is preserved. The configuration of these elements changes over time. This establishes an analogy with an ordinary isolated system, where there is chaotization, for example, of gas molecules in a closed vessel and an increase in entropy. This establishes a correspondence with the processes obeying the second law of of thermodynamics.

In principle, the structural distribution function can be found (and experimentally, too) by considering its change on large timescales; it is necessary to impose a spatial grid with a suitable choice of cells on the distribution of these elements in the biosystem and compare their position at different moments.

The structural distribution function makes it possible to determine changes in the “ideal structure” of an organism due to metabolic processes. In detail, the structural distribution function should be described similarly to the ordinary distribution function, perhaps for greater characteristic timescales. The distribution function *f_T_* (*t_T_*, *x_T_*, *e_T_*) describes the density of elements in a small vicinity of time point *t_T_*, coordinate *x_T_*, and energy *e_T_* (note that the scale of this vicinity of time may be sufficiently greater than a scale of metabolic time *τ*).

At large timescales, the system can be considered “closed” (in the sense of structuralism), and only the time *T* of structural degradation (aging) is significant. It does not matter which molecule as a result of metabolism will be in a given place within the structure. In other words, the biological structure itself is a closed object. We are interested in the density of the elements at given points; this density changes stochastically. This leads to thermodynamic equilibrium.

The relaxation process can be described in some local region of the system under consideration. For such a “closed” system, we represent this process similarly to the process of spatially uniform relaxation:(4)$$\frac{\partial f_{T}}{\partial t_{T}}=\frac{1}{T}\left(f_{MT}-f_{T}\right)$$
where *f_MT_* is the equilibrium “structural distribution function”. This equilibrium distribution is similar to the ordinary equilibrium distribution, but perhaps only for more large scales.

When these elements are replaced by other elements, i.e., when implementing a kind of “metabolism” on the timescale *T*, the growth of entropy may slow down or even be suspended. This implies that in order to overcome degradation, one can try to make the system open on a large timescale, so that it is possible to restore the structural distribution function at an “ideal age”. In essence, a new problem is posed: is it possible to define a stationary process in which the “exchange of elements” takes place on a timescale *T*?

The replacement of elements, e.g., molecules, biochemical components in metabolism, cells, etc. disrupts the structure. We consider the relaxation process according to Equation (4). In spatially uniform relaxation, the entropy can only increase (*H*-theorem). In terms of structure, this means that for large timescales, we are not interested in the metabolic processes of small timescales, and in this sense, the biological structure is closed. Thus, we can study slow degradation processes in a closed (or quasi-closed) system. This system tends irreversibly to equilibrium with increasing entropy because of the “thermal noise” inherent in all metabolic processes, or in biological terms, because of errors. As for anabolic and catabolic processes, their differentiation may take place at the next stage of our study; for now, we include them in the general metabolic process in the broad sense, constantly disturbing and only partially restoring its stationary structure.

In a simple case, we can consider only the dependence of the distributions *p*(*x*) and *q*(*x*) on the physical coordinates *x* at different points in time. Kullback–Leibler entropy is a suitable tool to describe this process. We can consider the density at different places in the organism, and this distribution is the subject of studies on the process of permanent changes in structure during metabolic interaction:(5)$$\xi \frac{\partial f_{T}}{\partial x_{T}}=\frac{1}{T}\left(f_{MT}-f_{T}\right)$$Here, in fact, a new problem is posed: is it possible to determine a stationary process in which the “metabolism” would occur on a timescale *T*?

When such elements are naturally or artificially replaced, it is possible to maintain the system in a stable state (for example, when cells in an organism are replaced). Note that, for example, the work [39] mentions the old analogy that compares the life of a cell and organism as a whole with the work of a machine: if all parts of a machine are replaced, will the machine be long-lived, or will the mutual arrangement of parts also change the connections between them, causing them to “wear out”? Such a process, which in one way or another is already being implemented “spontaneously”, is associated with the replacement of various organs, cell systems, or individual cells; in terms of structural distribution, it can be interpreted as requiring the replacement of the mentioned elements. As for the neurosystem, it seems that we can talk about the simultaneous replacement of a small number of neurons in order to keep the “whole” unchanged, e.g., the human personality, when the brain cells are replaced. The issue of the speed of such a replacement is a particularly complicated problem. The process of permanent cell replacement in a biosystem, leading to rejuvenation, consists of a coordinated rate of such a replacement, which seems to be accomplished by some animal species such as naked shrews (*Heterocephalus glaber*).

Different means for “opening up” biosystems can be considered; examples include the pathway of cell immortalization using telomerase expression induced by gene insertion, senile cell replacement, a generally constant stream of cell replacement with new stem cells, and organ transplantation. New opportunities arise in the creation of induced pluripotent stem cells, for example, by reprogramming the somatic cells of a given organism. Other approaches are also possible. Staged modelling of regeneration processes is being studied (see, for example, [40]). The steady state in large timescales of biosystems can be maintained using the artificially induced “metabolism on large timescales”; in particular, the following question can be posed: is the “genetically individual food” diet capable of influencing the above processes?

Genetically individualized nutrition (personalized nutrition) is associated with some of the special diets that exist today. Nutrigenetics and nutrigenomics are exploring new directions in the relationship between genomes and human nutrition. They are mainly aimed at overcoming certain diseases (see, for example, [41,42,43]).

However, we do not refer only to this trend. It is assumed that in future technologies we will be able to create food with elements subtly correlated with the individual genetic characteristics of the organism. Specially designed components of genetically oriented food will be able to restore stable metabolic conditions. Perhaps this will lead to a permanent restoration of low-entropy distribution functions in our terms.

We can also introduce into consideration the notion of the existence of direct and feedback connections in the system, which determine its behavior and responses to external influences. In this case, we approach a cybernetic description of the system. It can be assumed that one of the basic concepts for establishing such a relationship is the concept of cooperativity. The measure of cooperativity as a property of the connectivity of individual parts of a biosystem through entropy was discussed above.

What does aging have in common in all organisms? Obviously, aging occurs after a series of irreversible changes in the organism. These changes can be caused by both internal (ontogenesis) and external factors. The organism tries to cope with dysfunction due to some external influences using evolutionary instructions (including using the multifunctionality of organs [44]). But the main thing that happens during aging is that during this final stage of ontogenesis, the interconnections between different parts of the integral living system are also disturbed. On the organismal level, it is primarily the links between organs and physiological systems that are disturbed. In a number of cases, there is a weakening of the interaction and, as a consequence, a violation of the correct responses of some parts of the body to certain influences. Because of this, the quality of physiological reactions decreases. At the same time, there can be wrong reactions (for example, a spasm occurs, because negative feedbacks are weakened). In this context, a variety of causes of aging on both cellular and organismal levels can be considered and classified, e.g., the reduction of tissue nutrition due to impaired blood circulation, or the degradation of tissues producing some enzyme. One way or another, there is a deterioration of the relationship between the parts of the whole system and its fragments, and so the organism’s [44] cooperon eventually collapses. It seems that the beginning of the degradation of this cooperon should be regarded as the beginning of aging.

The concept and ideas of cooperativity are discussed in various fields of natural science. In physiology, the organism is viewed as a complex cooperative system. A feature of cooperative systems is that relatively independent components of such systems are functionally combined, and the functioning of each component is impossible without the existence and functioning of the other components. On the other hand, each component creates conditions for the functioning of other components. There can be many specific options for this kind of mutual assistance. It is difficult to indicate in general terms the specific mechanisms of cooperative interaction, which may be different in each case. As the organism ages, the cooperative interaction of its parts decreases and is disturbed, which eventually leads to the disintegration of the whole system and the death of the organism. The notion of cooperation (and the cooperon) allows us to apply combinatorial entropy, the value of which increases as the cooperative connection between the different parts of the organism decreases. In fact, it can also be introduced as a measure for assessing the age of an organism.

Of course, the idea of the destruction of the cooperativity of the organism system does not allow for the identification of a single “root cause” of the aging of the organism as a whole. The prolonged deterioration of the functioning of one of its parts, caused by some specific reasons, eventually leads to the deterioration of the functioning of its other dependent parts. Then, according to the domino principle, the functioning of the system as a whole is disturbed, and its cooperativity is gradually destroyed.

Revealing the general regularities and principles of ontogenesis that eventually lead to the “collapse” of the cooperon (i.e., the organism’s death) will be the goal of our further research.

## 4. Conclusions

In this work, we have considered various approaches to the problem of aging, which are united by one focus: the application of the concepts of statistical thermodynamics to the assessment of cell and organ degradation. In the future, we plan to develop these approaches.

The considered variants of the information theory of aging focus attention on the different types of information characterizing living objects. Thus, in A.V. Karnaukhov’s group, the emphasis is placed on genetic information, whereas in D. Sinclair’s group, it is placed on epigenetic information. Nevertheless, we believe that the general principle of information degradation during aging allows these variants to be considered as parts of a unified informational theory of aging. In addition, we believe that this theory can and should be supplemented with sections considering other types of information, e.g., information characterizing the phenotype of a living being.

In this work, we developed an approach that allows for the description of various types of information in living objects in a unified way, within a universal formalism; we also considered a simple illustrative example that shows the efficiency of such formalism. In the future, we plan to develop this approach in order to achieve understanding of the fundamental mechanisms of the functioning of living matter.

## Figures and Tables

**Figure 1 entropy-25-01067-f001:**
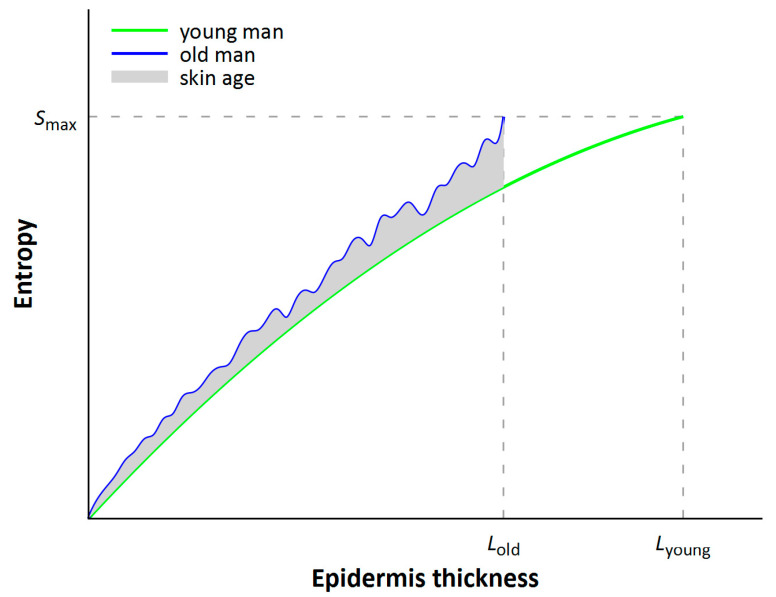
Schematic representation of cell entropy dependence on their coordinates in epidermis thickness for two ages of an organism. The zero mark corresponds to the greatest depth of cell location. In young and elderly people, the skin (on average) has different thicknesses, *L*_young_ and *L*_old_, respectively, with fluctuations in cell entropy increasing with age; the maximum entropy corresponds to a dead cell. The Python code for the production of this figure is outlined in the Appendix A.

**Figure 2 entropy-25-01067-f002:**
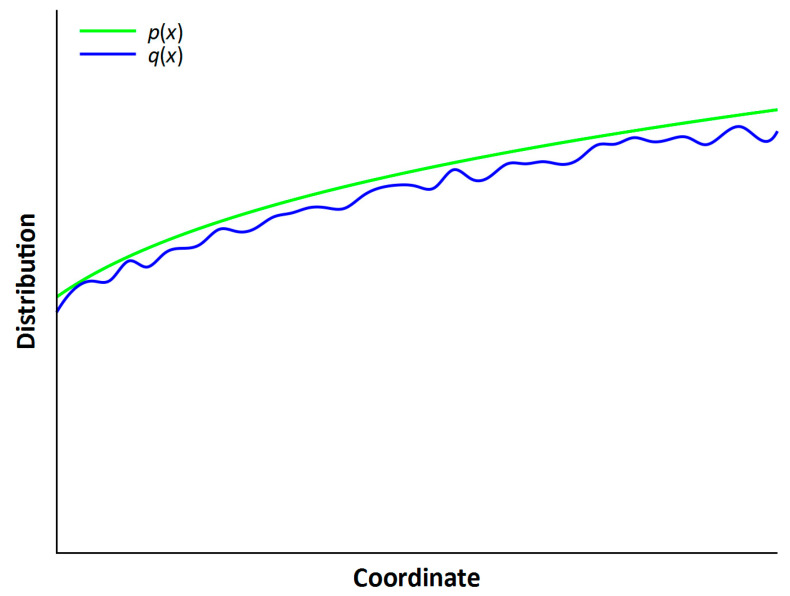
The distribution of the spatial density of cells and its change with time. The Python code for the production of this figure is outlined in the Appendix A.

**Figure 3 entropy-25-01067-f003:**
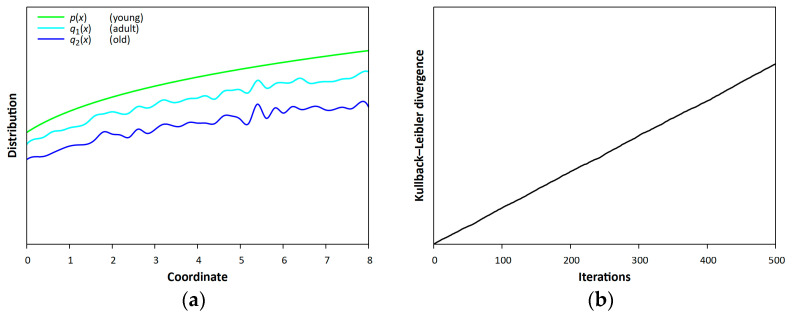
Changes in the skin structure with age along an arbitrary line (in conventional units), and the calculated corresponding Kullback–Leibler entropy. (**a**) Profiles *q*_1_(*x*) and *q*_2_(*x*) are two broken lines (after a number of iterations); (**b**) the change in Kullback–Leibler entropy with time, along the *x*-axis, after a number of iterations. The Python code for the production of this figure is outlined in the Appendix A.

## Data Availability

No new data were created or analyzed in this study.

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
