# Peer review of "The Degradation and Aging of Biological Systems as a Process of Information Loss and Entropy Increase"

_entropy, 2023, doi:10.3390/e25071067_

Round 1
Reviewer 1 Report
The manuscript titled " Degradation and Aging of Biological Systems As a Process of Information Loss and Entropy Increase " Overall, the text provides a general overview of the concepts and ideas related to the study of degradation, aging, and maintenance of biosystems, but clarifications on some of the mentioned concepts and their relationships is needed:
1. The concept of a "structural distribution function" introduced in the manuscript requires further elaboration. While the text provides a qualitative explanation, it lacks a precise definition and probably mathematical representation of this function. It would be beneficial to explicitly define the structural distribution function, including its mathematical formulation if available, to enhance clarity and understanding of its role in describing changes in the configuration of biological elements over time.
2. The relationship between the replacement of elements (cells, organs) and the growth of entropy needs to be clarified. The text suggests that the growth of entropy may slow down or be suspended when elements are replaced, but the causal relationship between element replacement and entropy is not adequately explained. It is recommended to provide a more explanation of how element replacement influences entropy, considering factors such as system configuration, energy distribution, and the principles of catabolic and anabolic processes.
3. The significance and implications of a "genetically individual food" diet in influencing the processes of aging and maintenance of biosystems are briefly mentioned but not sufficiently expanded upon. Although the text poses the question, it lacks further information or discussion on this topic. To address this, it is suggested to provide a more comprehensive or more references.
4. The concept of cooperativity and its relationship to entropy and aging are mentioned in the manuscript, but its precise definition and role within the study's context require further explanation. The text refers to cooperativity as a property of connectivity between individual parts of a biosystem through entropy, but the specific implications and underlying mechanisms remain unclear. To address this, it is recommended to provide a more explanation of cooperativity, and its role in the context of entropy and aging, to support the claims made.
Author Response
Responses to Reviewer 1
The authors express their gratitude to the reviewer for his very helpful comments and questions, the answers to which made it possible to improve the text and connect the parts of the paper.
- The concept of a "structural distribution function" introduced in the manuscript requires further elaboration. While the text provides a qualitative explanation, it lacks a precise definition and probably mathematical representation of this function. It would be beneficial to explicitly define the structural distribution function, including its mathematical formulation if available, to enhance clarity and understanding of its role in describing changes in the configuration of biological elements over time.
The response to this question is linked to the response to the next (2.) question. The structural distribution function allows us to determine changes in the “ideal structure” of an organism due to metabolic processes. In detail the structural distribution function should be described analogously to the ordinary distribution function maybe for greater characteristic scales of time. The distribution function fT (tT, xT, eT) describes the density of elements in a small vicinity of a point of time tT coordinate xT and energy eT (note that a scale of this vicinity of time could be sufficiently greater than a scale of metabolic time t).
On a large time scale, the system can be considered as “closed” (in the sense of structurality), only the time T of structural degradation (aging) is significant. It does not matter which molecule as a result of metabolism will be at a given place in the structure. In other words, the biological structure itself is a closed object. We are interested in the density at given points of the elements, this density changes stochastically. This leads to thermodynamic equilibrium.
The relaxation process can be described in some local region of the system under consideration. For such a “closed” system we represent this process similar to the process of spatially uniform relaxation
, (*)
where fTM is the equilibrium “structural distribution function”. This equilibrium distribution is similar to the ordinary equilibrium distribution maybe only for more large scales.
- The relationship between the replacement of elements (cells, organs) and the growth of entropy needs to be clarified. The text suggests that the growth of entropy may slow down or be suspended when elements are replaced, but the causal relationship between element replacement and entropy is not adequately explained. It is recommended to provide a more explanation of how element replacement influences entropy, considering factors such as system configuration, energy distribution, and the principles of catabolic and anabolic processes.
The response to this question is linked to the response to the previous (1.) question. The replacement of elements, e.g. moleculs, biochemical components in metabolism, cells, etc disturbs the structure. The process of the relaxation according to Eq. (*) is considered. In the spatially-uniform relaxation the entropy can be only increases (the H-theorem). From the point of view of structure, this means that for large time scales we are not interested in metabolic processes of small time scales, and in this sense, the biological structure is closed. Thus, we can study slow degradation processes in a closed (or quasi-closed) system. This system irreversibly tends to equilibrium with an increase in entropy due to the "thermal noise" inherent in all metabolic processes or, in biological terms, due to errors. (As for anabolic and catabolic processes, their differentiation could take place at the next stage of our study, while we include them in the general metabolic process in a broad sense, constantly disturbing and only partially restoring its stationary structure).
In the simple case we can consider only dependence on the physical coordinates x as for distributions p (x) and q(x) in different time moments. Kullback-Leibler entropy is a fit instrument for describing the process. We can consider the density in the different places of the organism and this distribution is a subject to study in the process of the permanent change of the structure in the metabolic interaction.
To overcome degradation, one can try to make the system open on a large time scale. Then, on these scales, the kinetic equation is stationary:
.
Here, in fact, a new problem is posed: is it possible to determine a stationary process in which the "metabolism" would occur on a time scale T. These propositions are qualitatively considered in Discussion of the paper.
- The significance and implications of a "genetically individual food" diet in influencing the processes of aging and maintenance of biosystems are briefly mentioned but not sufficiently expanded upon. Although the text poses the question, it lacks further information or discussion on this topic. To address this, it is suggested to provide a more comprehensive or more references.
Genetically individual nutrition (personalized nutrition) is a proposal associated with some of today's specific special diets. In nutrigenetics and nutrigenomics, new directions in the correlation of genomes and human nutrition are being explored. Which are aimed mainly at overcoming certain diseases. See for example:
- M. Ordovas, V. Mooser. Nutrigenomics and nutrigenetics. Curr. Opin. Lipidol. 2004. Apr;15(2):101-8.
- Ramos-Lopez et al. Guide for Current Nutrigenetic, Nutrigenomic, and Nutriepigenetic Approaches for Precision Nutrition Involving the Prevention and Management of Chronic Diseases Associated with Obesity. J Nutrigenet Nutrigenomics. 2017. 10. P. 43–62.
Lynnette R. Ferguson. Nutrigenomics and Nutrigenetics in Functional Foods and Personalized Nutrition. CRC Press. Taylor & Francis Group. London, New York. 2014.
But we have in mind not only this trend. It is assumed that in the technologies of the future, we will be able to create food with elements that are finely correlated with the individual genetic characteristics of the organism. Specially designed components of genetically oriented food will be able to restore stable metabolic conditions. Perhaps this will lead to a constant recovery of distribution functions with low entropy in our terms.
- The concept of cooperativity and its relationship to entropy and aging are mentioned in the manuscript, but its precise definition and role within the study's context require further explanation. The text refers to cooperativity as a property of connectivity between individual parts of a biosystem through entropy, but the specific implications and underlying mechanisms remain unclear. To address this, it is recommended to provide a more explanation of cooperativity, and its role in the context of entropy and aging, to support the claims made.
The concept of cooperativity and the ideas of cooperativity are discussed in various fields of natural science. In physiology, the organism is discussed as a complex cooperative system. A feature of any cooperative systems is that the relatively independent components of such systems are functionally combined, and the functioning of each of the components is impossible without the existence and functioning of other components. On the other hand, each component creates conditions for the functioning of other components. There can be many specific options for this kind of mutual assistance. It is difficult to indicate in a general way the specific mechanisms of cooperative interaction, which in each case may be different. With the aging of the organism, the cooperative interaction of the parts of the organism is disturbed and reduced, which ultimately leads to the collapse of the entire system and the death of the organism. The notion of cooperation (and cooperon) allows us to apply the combinatorial entropy which increases its value when the cooperative connection between different parts of the organism decreases. In fact it can also be introduce a measure to estimate the age of the organism.
We have changed the relevant places in the article and marked them in yellow.

Reviewer 2 Report
Comments on the work entitled: “Degradation and Aging of Biological Systems As a Process of Information Loss and Entropy Increase”.
The work addresses a topic of special interest for biomedical sciences, where the aging of biological systems remains an unknown, intriguing and challenging topic that deserve extensive discussion.
The work addresses the issue of aging from the statistical thermodynamics of non-equilibrium and focuses his attention on genetic and epigenetic aspects from information theory. It itself can be considered an extension of the previous works of the authors (Aristov, V.V.; Buchelnikov, A.S.; Nechipurenko, Yu.D. The Use of the Statistical Entropy in Some New Approaches for the Description of Biosystems. Entropy 2022, 24, 172, doi:10.3390/e24020172
Aristov, V.V.; Karnaukhov, A.V.; Levchenko, V.F. Nechipurenko; Yu.D. Entropy and Information in the Description of Bio-systems. Biophysics 2022, 67, 593–599, doi:10.1134/S0006350922040029).
We believe that a paper dealing with aging processes from thermodynamics is always welcome and fits the profile of the special issue of the magazine. It also has sufficient merits to be published, once the following recommendations and comments are taken into account.
Recommendations:
1. The references that appear below should be included and commented on, related to the subject matter and the formalism used.
Balmer, R.T. Entropy and Aging in Biological Systems. Chem. Eng. Commun. 1982, 17, 171–181.
Miquel, J.; Economos, A.C.; Johnson, J.E., Jr. A Systems Analysis—Thermodynamic View of Cellular and Organismic Aging, Aging and Cell Function; Springer: New York, NY, USA, 1984.
Aoki, I. Entropy principle for human development, growth and aging. J. Theor. Biol. 1991, 150, 215–223.
Gladyshev, G.P. The thermodynamic theory of evolution and aging. Adv. Gerontol. 2014, 4, 109–118.
Betancourt-Mar, J.A.; Mansilla, R.; Cocho, G.; Nieto-Villar, J.M. On the relationship between aging & cancer. MOJ Gerontol. Ger. 2018, 3, 163–168.
Nieto-Villar, J.M.; Mansilla, R. Longevity, Aging and Cancer: Thermodynamics and Complexity. Foundations 2022, 2, 664–680. https://doi.org/10.3390/foundations2030045
Moldakozhayev, A., & Gladyshev, V. N. (2023). Metabolism, homeostasis, and aging. Trends in Endocrinology & Metabolism.
Medvedev, Z.A. An attempt at a rational classification of theories of ageing. Biol. Rev. 1990, 65, 375–398.
Ghosh, C.; De, A. Basics of aging theories and disease-related aging-an overview. Pharma Tutor 2017, 5, 16–23.
Zotin, A.I. Thermodynamic Principles and Reaction of Organisms; Nauka: Moscow, Nauka, 1988.
Lucia, U.; Grisolia, G. Second law efficiency for living cells. Front. Biosci. 2017, 9, 270–275.
Khinchin A.I., Mathematical Foundations of Information Theory, Dover Publications, Inc., New York, 1957.
Klimontovich Yu. L., Statistical Physics, Nauka, Moscow, 1982; Harwood Academic Publishers, New York, 1986.
2. In the Abstract appears the sentences (lines 20,21): … Maintaining a biosystem in a time-stablestate requires a constant inflow of negative entropy (negentropy)…, but it is not commented on in the body of work.
Comments:
1. Fig. 1 shows …a schematic representation of the cell entropy dependence on their coordinates in the epidermis thickness…, (what entropy is being talked about? as the authors themselves state (line 132): … there are various definitions of entropy…; how was it evaluated?
2. In Eq. (2) how do you evaluate, Sold −Syoung ?
3. The authors introduce the relative entropy or Kullback-Leibler divergence, Eq. (3). What advantage does it offer? Although it is roughly estimated (lines 151, 152), it would be recommendable that the detailed calculation appears in the Appendix.
4. On the one hand, it is known that biological systems self-organize out of thermodynamic equilibrium due to the existence of bifurcations, which leads to the instability of stationary states (see, for example: Nicolis G. & Nicolis C. , Foundations of Complex Systems. Nonlinear Dynamics. Statistical Physics. Information and Prediction, World Scientific Publishing Co. Pte. Ltd, Singapore, 2007). On the other hand, it is known that the existence of a bifurcation compromises the scope of Eq. (3) limiting it only to stable stationary states, which is reflected in the change of shape of the stationary probability distribution. Could the authors offer an explanation about the scope and limitations of the relative entropy, Eq.(3) for these purposes?
Recall that the in the Abstract appears the statement (lines 20,21): … Maintaining a biosystem in a time-stable state requires a constant inflow of negative entropy (negentropy)…,
5. The authors state (184-186): … Our approach allows us to introduce a definition of a measure of aging of a biosystem. It is possible to quantify the age of the system using the entropy value calculated by Eq. (2) or Eq. (3); this will make it possible to estimate the biological age of the system..., can the authors illustrate it with a concrete example?
Author Response
Responses to Reviewer 2
The authors thank the referee for his very helpful comments, recommendations and suggestions, which allowed us to expand the list of references by adding several important works.
- Fig. 1 shows …a schematic representation of the cell entropy dependence on their coordinates in the epidermis thickness…, (what entropy is being talked about? as the authors themselves state (line 132): … there are various definitions of entropy…; how was it evaluated?
Here, in Fig. 1, the statistical entropy can be applied using the distribution function in the kinetic formulation. This relates to our work [3], where we can estimate the local statistical entropy and find that during aging, relaxation processes prevail over advection (transport) processes.
Now we restrict ourselves to a theoretical consideration of the problem, but in principle one can resort to an experimental description of nonequilibrium distributions of. For a rougher estimation method, the Kullback-Leibler entropy can be used, see Fig. 2. This apparatus is used later for a simple comparison of the distributions p(x) and q(x).
- In Eq. (2) how do you evaluate, Sold−Syoung?
This difference can be calculated using the distribution function for the kinetic formulation as mentioned above, or more roughly using (3) according to Kullback-Leibler entropy.
- The authors introduce the relative entropy or Kullback-Leibler divergence, Eq. (3). What advantage does it offer? Although it is roughly estimated (lines 151, 152), it would be recommendable that the detailed calculation appears in the Appendix.
- On the one hand, it is known that biological systems self-organize out of thermodynamic equilibrium due to the existence of bifurcations, which leads to the instability of stationary states (see, for example: Nicolis G. & Nicolis C. , Foundations of Complex Systems. Nonlinear Dynamics. Statistical Physics. Information and Prediction, World Scientific Publishing Co. Pte. Ltd, Singapore, 2007). On the other hand, it is known that the existence of a bifurcation compromises the scope of Eq. (3) limiting it only to stable stationary states, which is reflected in the change of shape of the stationary probability distribution. Could the authors offer an explanation about the scope and limitations of the relative entropy, Eq.(3) for these purposes?
Recall that the in the Abstract appears the statement (lines 20,21): … Maintaining a biosystem in a time-stable state requires a constant inflow of negative entropy (negentropy)…,
In our kinetic model [3], a biological system is considered as a nonequilibrium structure created as a result of the action of two factors: biochemical reactions and advection (transport). To ensure a stationary bio-state. This is also connected with the continuous flow of negentropy. From our point of view, the stationary state is maintained by continuous metabolism, interruption of one of the mentioned factors (for example, advection) leads to local equilibrium. Aging is a similar but very slow process. We study a biological system in quasi-steady, and p(x), q(x) are two such states.
- The authors state (184-186): … Our approach allows us to introduce a definition of a measure of aging of a biosystem. It is possible to quantify the age of the system using the entropy value calculated by Eq. (2) or Eq. (3); this will make it possible to estimate the biological age of the system..., can the authors illustrate it with a concrete example?
For illustration, the Kullback-Leibler entropy for the epidermis can be used, our estimates are theoretical, but we hope that experimental tests can be done in the future. Age can be related to the value of the Kullback-Leibler entropy. Suppose three divergence values can be attributed to three different ages and the appropriate estimates can be made.
Chronic inflammation is one of the main biological markers of aging and elements of the pathogenesis of age-related diseases. In this case, the processes of the immune response demonstrate misalignment with age. This procees can be intuitively associated with an increase in entropy in a biological system. But it is also possible to give some quantitative estimate of the increase in entropy, if the concept of combinatorial entropy is used correctly (see Appendix).There is also an increase in autoimmune response, one of the causes of which is the accumulation of genetic errors in cellular genomes, which reflects the degradation of genetic information of organism. On the example of such a phenomenon as chronic inflammation. We plan in the future to build a specific simulation model in which the processes of aging, the growth of the entropy of the system and the degradation of genetic information will be interconnected.
Another example can be attributed to inflammation, see, e. g.:
- Sanada et al. Source of Chronic Inflammation in Aging. Front. Cardiovasc. Med., 22 February 2018. | https://doi.org/10.3389/fcvm.2018.00012
We have changed the relevant places in the article and marked them in yellow.

Round 2
Reviewer 1 Report
I am delighted to inform you that I am extremely satisfied with the revisions made to the manuscript titled "Degradation and Aging of Biological Systems As a Process of Information Loss and Entropy Increase."
Reviewer 2 Report
The changes made by the authors are sufficient. I recommend the publication of the new version.